# Evaluation of the Wind Field and Deposition Effect of a Novel Air-Assisted Strawberry Sprayer

**Shaoqing Xu** [1,2], **Yuru Feng** [2,3], **Leng Han** [1,2], **Xiangkai Ran** [1,2], **Yuan Zhong** [1,2], **Ye Jin** [1,2] **and Jianli Song** [1,2,*]

1   College of Science, China Agricultural University, Beijing 100193, China
2   Centre for Chemicals Application Technology, China Agricultural University, Beijing 100193, China
3   Sanya Institute of China Agricultural University, Sanya 572025, China
*   Correspondence: 13520162346@163.com

**Abstract:** Strawberry is a widely cultivated cash crop in China. In order to control pests and diseases on strawberries, there must be sufficient deposits on the abaxial surfaces of the leaves. Air-assisted technology can effectively increase the deposition on the abaxial surfaces of the leaves; however, most air-assisted equipment is not suitable for application due to the pattern of strawberry planting. Therefore, a novel air-assisted strawberry sprayer was developed, the static and dynamic wind fields were measured using a 3D anemometer, and the effectiveness of the application at different spray angles and wind speeds was evaluated. In addition, a comparison of the deposition effect in the strawberry canopy between the air-assisted strawberry sprayer, knapsack sprayer, and spray gun was conducted. The results showed that in the static wind field test, a difference between the center and edge wind fields was obtained, which was correlated with the distance and the outlet wind speed. In the dynamic wind field test, the wind field was found to be rolling backward during the movement, and an inward vortex was obtained. In the field, the data showed that a spray angle of $30°$ and a wind speed of $16 \text{ m·s}^{-1}$ had the best deposition on the abaxial surface, with a coverage of 36.5% and 38.3% in the upper canopy and 6.2% and 7.9% in the lower canopy, respectively. Moreover, the air-assisted strawberry sprayer was found to have a higher deposition efficiency on abaxial surfaces than the knapsack sprayer and spray gun at a lower spray volume, the values of which in the upper and lower canopies were 42.8% and 29.7%, respectively. In conclusion, the air-assisted strawberry sprayer has the potential for the crop protection of greenhouse strawberries, and more evaluations are needed to improve the sprayer in the future.

**Keywords:** strawberry sprayer; air-assisted spray; greenhouse; wind field; deposition

## 1. Introduction

Strawberry is a widely cultivated cash crop worldwide [1,2] due to its high economic benefits and rich nutritional value [3,4]. According to the Food and Agriculture Organization of the United Nations (FAO) [5], strawberries were grown on 383,600 hectares worldwide, with a total production of 889,400 tons in 2020. China is one of the largest strawberry-growing countries [6], accounting for 33.06% and 37.65% of the world's strawberry cultivation and production in 2020. Growing strawberries has become a major source of income for farmers in many areas of China [7]. Since strawberry is sensitive to temperature and illumination [8–10], it is mainly cultivated in greenhouses with high monopolies in northern China [11,12]. However, the occurrence of pests and diseases can affect the yield and economic value of strawberries [13,14].

As the most common disease of strawberries, powdery mildew can reduce strawberry yield by 10–20% and up to 50% in severe cases [15,16]. Dang (2020) found that powdery mildew occurred mainly on the abaxial surfaces of strawberry leaves [17], where the pesticide deposition is insufficient, making it difficult to eradicate powdery mildew [18]. Therefore, it is crucial to improve the deposition efficiency on the abaxial surfaces of strawberry leaves.

Air-assisted spraying is a widely used application technique in orchards. Pesticide droplets are delivered to the target through airflow, which increases the penetration of droplets in the canopy compared with non-air-assisted sprayers [19,20]. Meanwhile, leaves can be disturbed by the airflow, thus improving the deposition of droplets on the abaxial surfaces of leaves. In previous studies, air-assisted spray has been used for tomato application in a greenhouse and obtained good results [21,22].

Strawberry is an herbaceous plant with short growth, and most air-assisted equipment is not suitable for this application; thus, the use of hand-held sprayers is still very common. Researchers have evaluated the effectiveness of handheld sprayer applications on strawberries, and the results showed that the deposition was not satisfactory [23]. In addition, the pesticide exposure of the applicator was high when using a handheld sprayer [24,25]. Therefore, researchers have developed different applicators to improve the efficiency of strawberry application. Vertical boom sprayers are the most common alternative. Ramos et al. (2000) developed a vertical boom sprayer and achieved a good effect [26]. Similarly, Braekman et al. (2010) proved that the deposition efficiency was higher than with a spray gun [27]. With the development of technology, automated equipment has been used in greenhouse strawberry applications [28]. Although these devices showed good performance, none of these studies evaluated the deposition effect on the abaxial surfaces of strawberry leaves, and the effectiveness of the prevention of powdery mildew cannot be guaranteed. In addition, most traditional greenhouses in northern China do not have the conditions to modify the automatic application equipment. Efficient equipment for strawberry applications in greenhouses in northern China is needed.

Therefore, we developed a hand-pushed air-assisted strawberry sprayer (ASS) with a duckbill air-assisted spray unit that can adapt to greenhouses in northern China. In this paper, we measured the wind field of the sprayer first, then evaluated its effect on the deposition in the strawberry canopy during the field application, and compared it with another two common applicators in a greenhouse.

## 2. Materials and Methods

### 2.1. Design of the Air-Assisted Strawberry Sprayer

We designed an air-assisted spray unit with a fan, duckbill hood, and nozzle (Figure 1). An axial fan was installed inside the duckbill hood, which had a diameter of 130 mm and was driven by a 12 V DC motor. The rotation speed could be adjusted with a potentiometer. The structure of the wind outlet was a duckbill, which could form a 60° fan-shaped airflow field during operation. The nozzle was mounted on the side of the hood, and the pressure range was 0–0.8 Mpa.

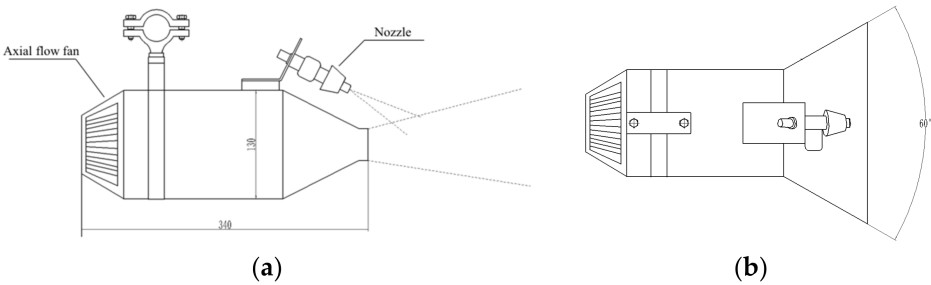

(**a**)　　　　　　　　　　　　　　　　(**b**)

**Figure 1.** Structure of the air-assisted spray unit: (**a**) side view of the air-assisted spray unit; (**b**) top view of the air-assisted spray unit.

The ASS consisted of an air-assisted spray unit, tank, base platform (including the power switch, pressure gauge, battery, and pump), and control panel (Figure 2). To facilitate its operation in tight spaces, two wheels were mounted on the sprayer; the front wheel was the main wheel, and the rear wheel was an auxiliary universal wheel. The spraying unit was fixed by fasteners with an adjustable height range of 0.3–1.7 m and an adjustable

angle range of 0–180°. In addition, two air-assisted spraying units could be controlled individually through the control panel below the handle position.

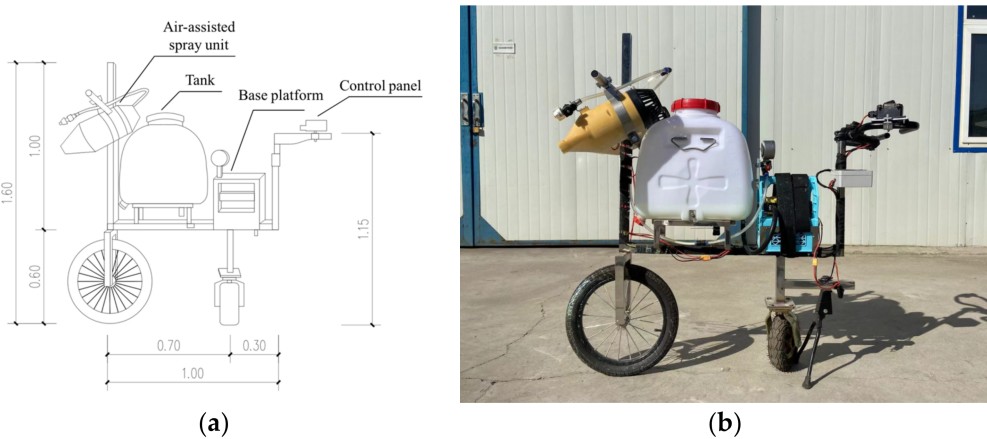

| (a) | (b) |

**Figure 2.** Structure of the ASS: (**a**) the design drawing of ASS; (**b**) the live picture of ASS.

### 2.2. The Wind Field of the Air-Assisted Spray Unit

In order to evaluate the wind field characteristics of the air supply unit, we measured the static as well as the dynamic wind fields of the air-assisted unit, and the tests were conducted indoors.

#### 2.2.1. Test of the Static Wind Field

As shown in Figure 3a, the wind delivery unit was fixed at a height of 1.5 m above the ground, and the wind field was measured by five WINDMASTER 3D anemometers (Gill Instruments Limited, UK), which were 1.1, 1.3, 1.5, 1.7, and 1.9 m above the ground, respectively. The 3D coordinates of the anemometer were shown in Figure 3b, the wind speed in the UV plane (windspeed), the wind direction in the UV plane (wind direction), and the wind speed in the W-axis (W_speed) could be obtained. The W-axis is perpendicular to the variable wind-assisted spray unit, and the N pointer of the anemometer was located on the backwind side. The fan was turned on at the beginning of the experiment, and data were collected when the fan was working steadily; each group of parameters was tested continuously for 30 s, and the acquisition frequency of the 3D anemometer was 10 Hz. In the test, we varied the outlet wind speed to 9, 12, and 16 m·s$^{-1}$. For each wind speed, the wind field of the horizontal distance of the wind delivery unit from the anemometer was tested at 0.3, 0.5, 0.7, and 0.9 m, respectively.

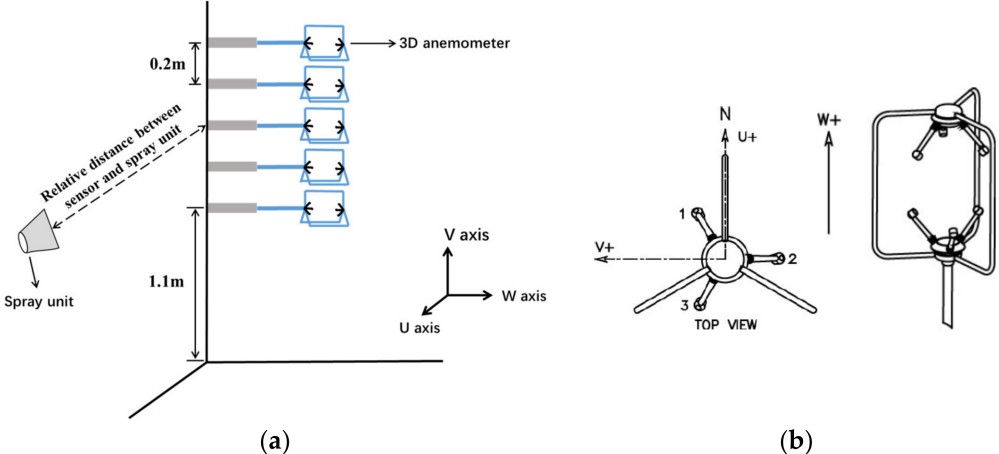

| (a) | (b) |

**Figure 3.** Diagram of the wind field test: (**a**) location of the air-assisted spray unit and 3D anemometers; (**b**) 3D coordinates of the anemometer.

When calculating the static wind speed, only the wind speed in the UV plane of the 3D anemometer was considered, and the wind speed was calculated as follows:

$$WS_n = \sum_{i=1}^{t \cdot f} WS_i / (t \cdot f) \tag{1}$$

where $WS_n$ is the wind speed of the particular test point, $t$ is the testing time of the wind field, and $f$ is the acquisition frequency of the 3D anemometer.

When processing the data, the locations of the sensors were redefined. The bottom-up vertical coordinates of the 5 sensors were −0.4, −0.2, 0, 0.2, and 0.4 m, based on the height of the air-assisted spray unit. Additionally, the distance between the sensor and the air outlet influenced the wind speed of the middle part (−0.2, 0.0 and 0.2 m on the y-axis) and the edge part (−0.4 and 0.4 m on the y-axis). To ensure the wind field could cover the whole plant in the application and reflect the change, the wind speed of the middle part was defined as the average wind speed of −0.2, 0.0, and 0.2 m, as well as the edge part, which was at −0.4 and 0.4 m; then, the difference between them was calculated.

### 2.2.2. Test of the Dynamic Wind Field

The test setup for the dynamic wind field was the same as that for the static wind field. The air-assisted spray unit was fixed to a crawler that could move at a uniform speed; the horizontal distance between the 3D anemometer and the air-assisted spray unit was 0.5 m. The wind speeds of the air outlet in the test were 9, 12, and 16 m·s$^{-1}$, and the speeds of the air-assisted spray unit were 0.8 and 1.2 m·s$^{-1}$, respectively.

When drawing, the output of the 3D anemometer was split. The arrows indicate the wind field on the UV plane, and the heat map indicates the wind field on the W-axis.

### 2.3. The Effect of the Air-Assisted Strawberry Sprayer with Different Operation Parameters

In order to investigate the effect of the different parameters and to screen for reasonable parameters for the sprayer operation, we conducted a trial in Jinliuhuan Agricultural Park, Changping District, Beijing, China. The period was from 14 December 2021, to 16 December 2021, and the strawberry was flowering (BBCH 65). In the experiment, the air speed of the air outlet was changed, as was the angle of the air-assisted spray unit (Table 1).

**Table 1.** The spray parameters of the ASS in the experiment.

| Group | Spray Unit Angle (°) | Wind Speed (m·s$^{-1}$) |
| --- | --- | --- |
| 1 | 30 | 9 |
| 2 | 30 | 12 |
| 3 | 30 | 16 |
| 4 | 50 | 12 |
| 5 | 70 | 12 |

The canopy was divided into six parts for every two strawberry plants: two parts in the vertical direction, where the upper layer was marked as "1" and the lower layer was "2", and three parts in the direction of the vertical ridge, marked as "A", "B", and "C". (Figure 4a). Water-sensitive paper (WSP, Syngenta Crop Protection AG, Basel, Switzerland) was arranged on the adaxial and abaxial surfaces of the leaves in each part. Five groups of strawberries (two strawberry plants per group) were arranged with WSP for each spary operation. After adjusting the air-assisted spray unit to the setting angle using an inclinometer (JingYan Instruments & Technology co., Ltd., China), the spraying operation was started. The route of the application is shown in Figure 4b. The nozzle used in the operation was a fan-shaped nozzle (Lechler ® LU110-015, Changzhou, China) with a spray pressure of 0.3 MPa, an atomization level of "fine", and an application volume of 225 L·ha$^{-1}$. After each spray operation, when the WSP was completely dried, it was collected into a card case, scanned at 600 dpi, and the coverage was analyzed using ImageJ

(National Institutes of Health, Bethesda, MD, USA). Figure 5 shows the field operation and the condition of the WSP after application. Three replications were performed for each set of parameters. And to eliminate the effect of plant differences on the experimental results, each replicate was operated in a different block.

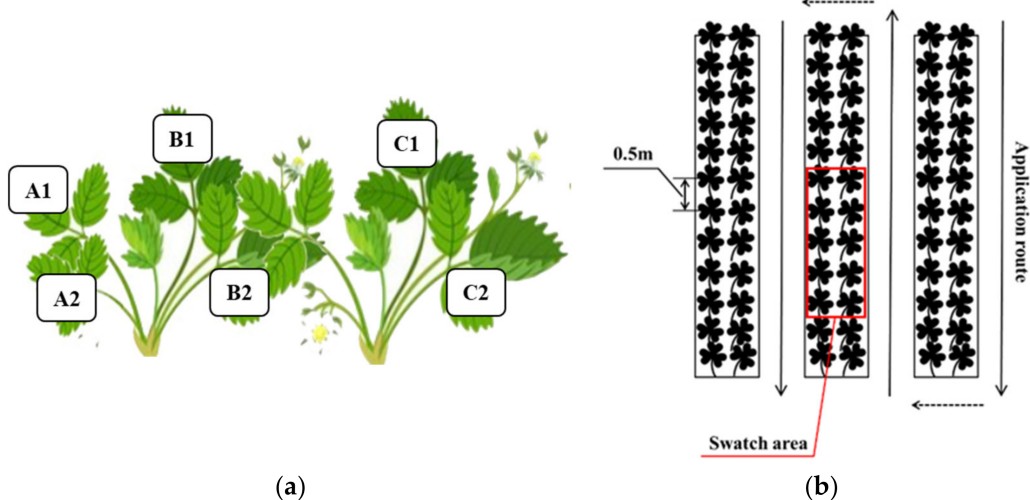

(**a**) (**b**)

**Figure 4.** Schematic diagram of the field test and canopy division: (**a**) division of the strawberry canopy; (**b**) experimental site and the route of application.

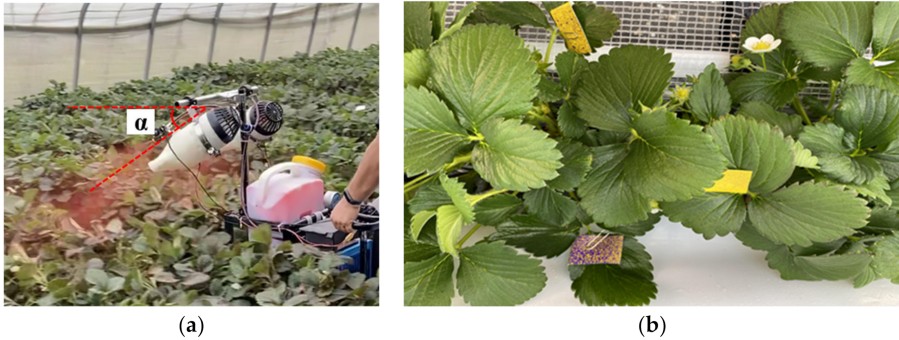

(**a**) (**b**)

**Figure 5.** Field operation and deposition effects of the ASS: (**a**) application in the greenhouse, where $\alpha$ is the angle of the air-assisted spray unit relative to the ground; (**b**) condition of the WSP after applying.

In the data processing, the deposition efficiency of the abaxial surface (*DEAS*) was used to express the deposition ability of the droplets on the abaxial surfaces of the leaves, which was calculated as follows:

$$DEAS = \frac{ABSC}{ADSC} \times 100\% \tag{2}$$

where *ABSC* is the coverage of the abaxial surface, and *ADSC* is the coverage of the adaxial surface.

*2.4. Comparative Spray Deposits by the Air-Assisted Strawberry Sprayer, Knapsack Sprayer and Spray Gun*

A knapsack sprayer (KS) and a spray gun (SG) are two common sprayers used in greenhouses. In order to evaluate the operational effectiveness of the strawberry wind-driven sprayers more comprehensively, we compared the deposits by ASS, KS (3WBD20L, Xuzhou Lanyi Plant Protection Equipment Co., Ltd., China, Figure 6a), and SG (3WH-40, Yifen Wanshan Technology Co., Ltd., China, Figure 6b). Additionally, the angle of the air-assisted spray unit on the ASS was 30° in the application, with a wind speed of 16 m·s$^{-1}$.

The nozzle type and application volume of the above three sprayers are shown in Table 2. The method for the WSP's placement and calculation of the coverage was the same as that in Section 2.3.

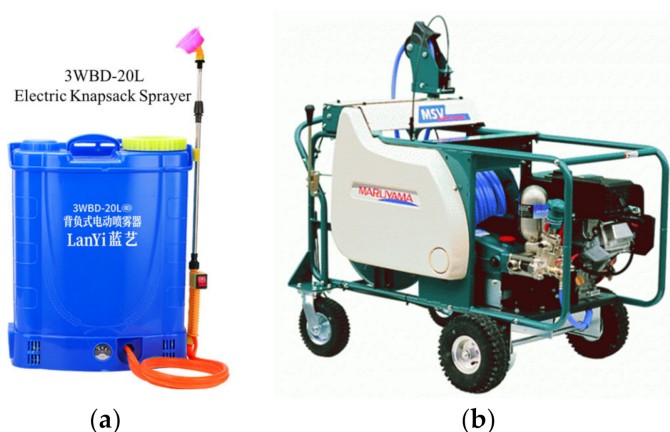

|        (**a**)        |        (**b**)        |

**Figure 6.** Two types of applicators for comparison: (**a**) knapsack sprayer; (**b**) spray gun.

**Table 2.** The nozzle and spray volume of the ASS, KS and SG.

| Applicator | Nozzle Type | Spray Volume (L·ha⁻¹) |
|---|---|---|
| ASS | Fan-shaped nozzle | 225 |
| KS | Cone nozzle | 540 |
| PB | Fan-shaped nozzle | 1022 |

## 3. Results and Discussion

### 3.1. Airflow Field of the Air-Assisted Spray Unit

#### 3.1.1. Static Wind Field

The result of the measurement of the static wind field is shown in Figure 7. Since the angle of the outlet was 60°, the wind field coverage increased with the increasing distance. More specifically, the farther away from the outlet, the smaller the difference between the edge wind speed and middle wind speed. Table 3 shows the specific values of the difference between the middle and the edge wind fields at different distances. Finally, at 0.9 m from the outlet, the edge wind speed was larger than the middle, with a difference of 0.14–0.32 m·s⁻¹. It should be noted that when the wind speed at the outlet was 9 m·s⁻¹, the edge wind speed was larger than the middle at a position of 0.7 m, which may have been caused by the fast spreading of the axial wind field. Based on our experience, the distance between the strawberry and the air-assisted spray unit was approximately 0.5 m during operation. In addition, the values were 2.1, 3.0 and 3.7 m·s⁻¹ in this position when the outlet wind speeds were 9, 12, and 16 m·s⁻¹, respectively.

**Table 3.** Difference between the middle and the edge wind fields at different distances.

| Distance between Sensor and Air Outlet (m) | Wind Speed (Difference between Middle and Edge Wind Speeds (m·s⁻¹)) | | |
|---|---|---|---|
|  | 9 m·s⁻¹ | 12 m·s⁻¹ | 16 m·s⁻¹ |
| 0.3 | 1.69 | 2.45 | 3.14 |
| 0.5 | 0.37 | 0.52 | 1.05 |
| 0.7 | −0.04 | 0.06 | 0.17 |
| 0.9 | −0.14 | −0.32 | −0.30 |

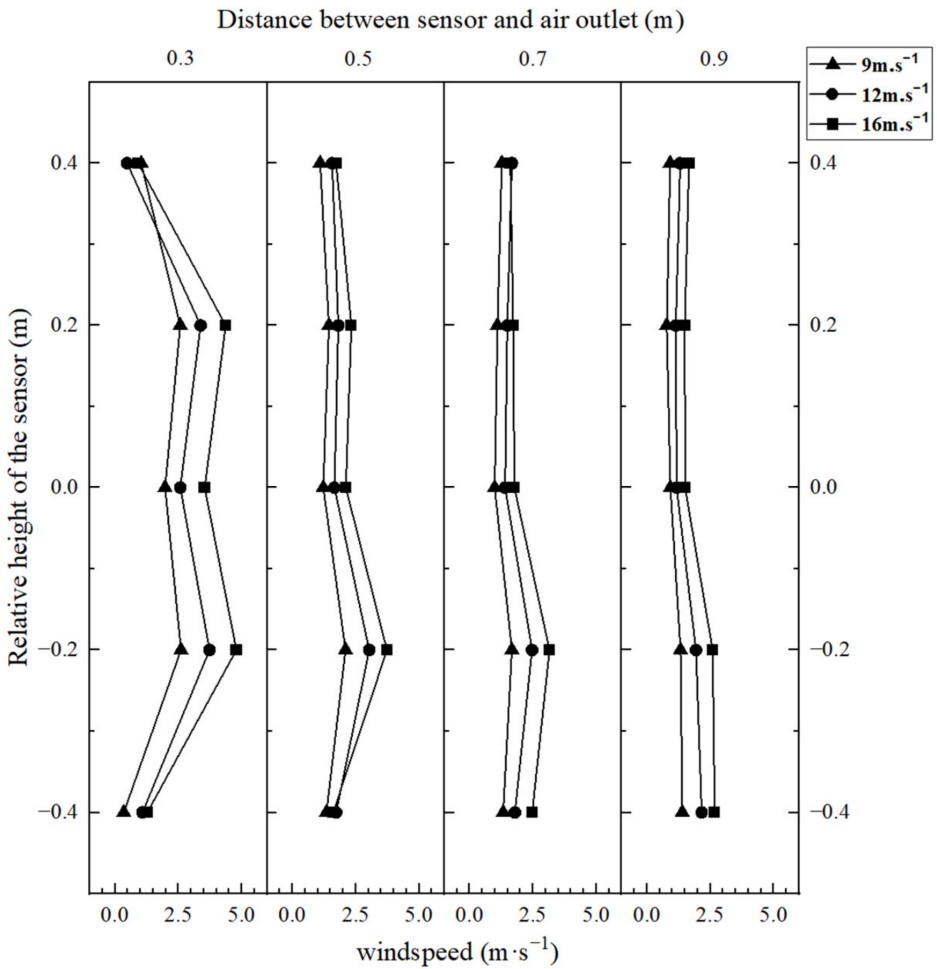

**Figure 7.** The static wind field of the air-assisted spray unit at different outlet wind speeds.

### 3.1.2. Dynamic Wind Field

We can observe from Figure 8 that when the air-assisted spray unit passed the sensor, the wind speed on the W-axis varied. The sensor in the middle part ($-0.2$, 0.0, and 0.2 m on the y-axis) can detect the wind speed in the positive direction of the W-axis because the spray unit has a forward speed, which causes the wind field to have a starting speed in the same direction. Additionally, W_speed increases as the forward speed increases, while the opposite W_speed was picked up by the sensors in the edge part ($-0.4$ and 0.4 m on the y-axis), where the W speed decreased as the travel speed increased (Figure 9). It can be deduced that as the air-assisted spray unit moved, the wind field farther from the center experienced a backward deflection, which caused the wind generated to curl in the opposite direction of the movement; it resembled the jet's trajectory in a cross-flow situation [29]. In addition, this phenomenon was positively correlated with the outlet wind speed and negatively correlated with the forward speed.

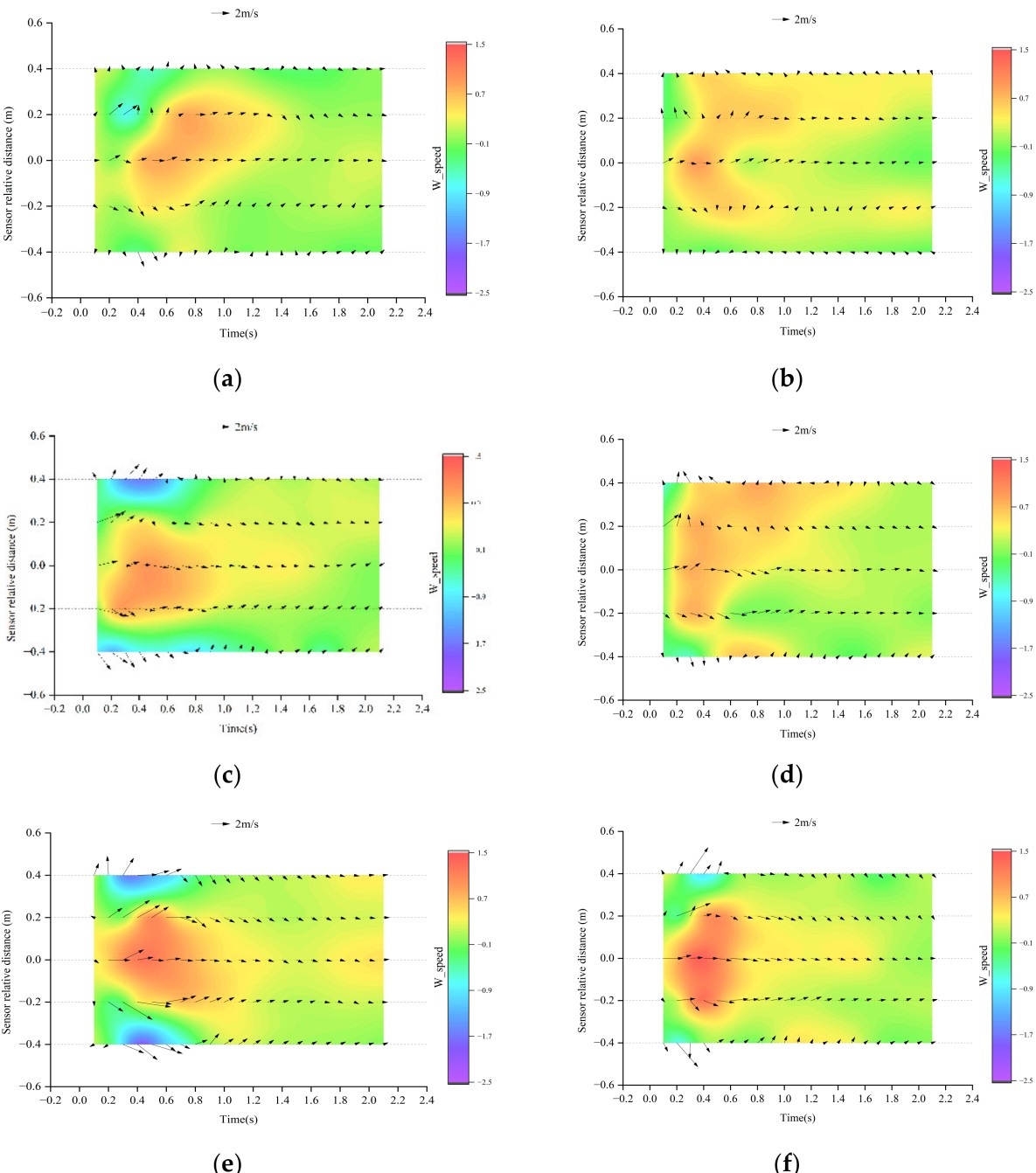

**Figure 8.** Dynamic wind field at different movement speeds and outlet wind speeds of the air-assisted spray unit: (**a**) movement speed of 0.8 m·s$^{-1}$ and outlet wind speed of 9 m·s$^{-1}$; (**b**) movement speed of 1.2 m·s$^{-1}$ and outlet wind speed of 9 m·s$^{-1}$; (**c**) movement speed of 0.8 m·s$^{-1}$ and outlet wind speed of 12 m·s$^{-1}$; (**d**) movement speed of 1.2 m·s$^{-1}$ and outlet wind speed of 12 m·s$^{-1}$; (**e**) movement speed of 0.8 m·s$^{-1}$ and outlet wind speed of 16 m·s$^{-1}$; (**f**) movement speed of 1.2 m·s$^{-1}$ and outlet wind speed of 16 m·s$^{-1}$.

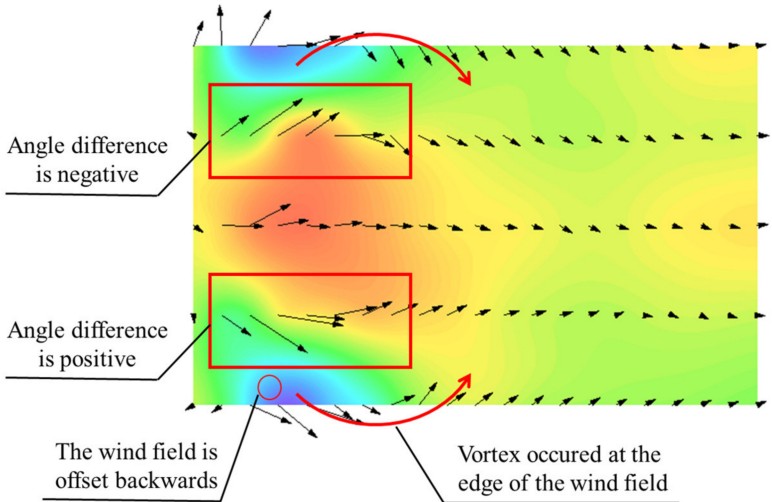

**Figure 9.** Offsets and vortices in the dynamic wind field.

In the UV plane, due to the jet angle, the wind field in the center was parallel to the U-axis, but the wind fields on either side were at an angle to the U-axis. The wind field initially dispersed to both sides with the movement of the wind-fed unit before contracting inward, with the contraction being most noticeable at approximately 0.8 s. As a consequence, we determined the angle between the wind field at 0.8 and 0.2 s at these two positions (Table 4). The result shows that the angular difference at −0.2 m was essentially positive, indicating that the wind direction at this location deflected clockwise with time while the wind direction at 0.2 m was the opposite; these two wind fields formed an inward vortex (Figure 9). Additionally, the largest vortex was created when the outlet wind speed was 16 m·s$^{-1}$, which was between 1.24 and 1.44 m·s$^{-1}$. Combined with the wind field of the W-axis, we can conclude that the vortex was generated by the previous backward-deflected wind field—when the outer wind field was deflected in the opposite direction, a vortex formed.

**Table 4.** Dynamic wind field of the UV plane at −0.2 and 0.2 m.

| Location | Forward Velocity (m·s$^{-1}$) | Outlet Wind Speed (m·s$^{-1}$) | Angle Difference (°) | Wind Speed (m·s$^{-1}$) |
|---|---|---|---|---|
| −0.2 m | 0.8 | 9 | 65 | 0.93 |
| | | 12 | 23 | 1.09 |
| | | 16 | 49 | 1.44 |
| | 1.2 | 9 | −91 | 0.28 |
| | | 12 | 35 | 1.29 |
| | | 16 | 31 | 1.38 |
| 0.2 m | 0.8 | 9 | −39 | 0.70 |
| | | 12 | −59 | 0.91 |
| | | 16 | −64 | 1.43 |
| | 1.2 | 9 | −92 | 0.82 |
| | | 12 | −155 | 0.57 |
| | | 16 | −48 | 1.24 |

In practice, the vortex can subject the strawberry leaf to the force of the wind from multiple directions, and the airflow pushes the leaf to increase the intensity of the turbulence in the canopy, therefore increasing the amount and coverage of the droplets deposited on the abaxial surfaces of the leaves.

### 3.2. Effect of the Spray Angle and Wind Speed on the Distribution of the Deposition in the Strawberry Canopy

3.2.1. Effect of the Spray Angle

Table 5 displays the coverage of the ASS at various spray angles, and the results show significant differences in the coverage at different strawberry canopy positions. For the upper canopy, the *DEAS* was 96% at a spray angle of 30°, with a high efficiency of the droplet deposition on the abaxial surface of the leaf. When the spray angle was adjusted to 50° and 70°, the *DEAS* values decreased to 60.8% and 29.2%, respectively, with a significant difference in the coverage on the adaxial and abaxial surfaces of the leaf. The variation pattern of the *DEAS* for the lower canopy was the same as the upper canopy. Although the coverage of the adaxial surfaces was greater than 30%, it was less than 10% on the abaxial surface, which was woefully inadequate and produced a significant difference. In pesticide application, a coverage of less than 10% may lead to ineffective control [30].

**Table 5.** Coverage (%) of the leaves and the *DEAS* values at different spray angles.

| Angles (°) | Leaves of Upper Canopy | | | Leaves of Lower Canopy | | |
| --- | --- | --- | --- | --- | --- | --- |
| | Adaxial Surface | Abaxial Surface | *DEAS* (%) | Adaxial Surface | Abaxial Surface | *DEAS* (%) |
| 30 | 38.0 cd | 36.5 cd | 96.0 | 35.5 cd | 6.2 a | 17.4 |
| 50 | 38.8 d | 23.6 bc | 60.8 | 32.4 cd | 5.4 a | 16.7 |
| 70 | 45.8 d | 13.4 ab | 29.2 | 32.4 cd | 4.1 a | 12.7 |

Note: Different letters indicate a significant difference in the $p < 0.05$ level.

The results suggest that the spray angle has an impact on the dispersion of the droplets in the canopy, which is similar to other studies [31–33]. The best deposition efficiency on the abaxial surfaces was at a spray angle of 30°, and as the angle increased to 50° and 70°, it became less effective. This might be because the leaf was forced down by the wind when the spray angle became larger, causing ineffective deposition on the abaxial surface. The research of Wu et al. (2021) found that when the angle between the leaves and the wind field changed, the aerodynamic response velocity also changed [34], which affected the distribution of the droplets deposited on the leaves; this may also account for the variation in the coverage in different parts of the canopy. Therefore, for managing diseases that frequently appear on the abaxial surfaces of leaves, a spray angle of 30° is more effective.

3.2.2. Effect of the Wind Speed

The coverage of the ASS at different wind speeds is shown in Table 6. Under the experimental circumstances, the coverage on the adaxial surfaces of the leaves was negatively correlated with the wind speed, while the abaxial surfaces of the leaves were opposite, which eventually caused an increase in the *DEAS*. For the upper canopy, there were significant differences in the coverage of the adaxial and abaxial surfaces of the leaves at a wind speed of 9 m·s$^{-1}$, and the *DEAS* was only 38.9%, while the deposition efficiency on the abaxial surfaces of the leaves was satisfactory at 12 and 16 m·s$^{-1}$, with *DEAS* values of 96.0% and 112.5%. The adaxial surfaces of the leaves were well deposited in the lower canopy, whereas the abaxial surfaces were less adequately covered. Even at a wind speed of 16 m·s$^{-1}$, the coverage of the abaxial surfaces was still less than 10%, despite the fact that the effectiveness of the deposition increases as the wind speed increases.

**Table 6.** Coverage (%) of the leaves and the *DEAS* values at different outlet wind speeds.

| Wind Speed (m·s⁻¹) | Leaves of Upper Canopy | | | Leaves of Lower Canopy | | |
|---|---|---|---|---|---|---|
| | Adaxial Surface | Abaxial Surface | *DEAS* (%) | Adaxial Surface | Abaxial Surface | *DEAS* (%) |
| 9 | 50.8 d | 19.8 ab | 38.9 | 35.7 bcd | 5.0 a | 13.9 |
| 12 | 38.0 cd | 36.5 bcd | 96.0 | 35.5 bcd | 6.2 a | 17.4 |
| 16 | 34.1 bcd | 38.3 cd | 112.5 | 30.5 bc | 7.9 a | 25.8 |

Note: Different letters indicate a significant difference in the $p < 0.05$ level.

The longer petiole of the strawberry makes the leaves more susceptible to airflow, and when the wind speed increases, the disruption of the blade becomes more intense, which accounts for the difference in the *DEAS* value. Combined with our analysis of the dynamic wind field in Section 3.1.2, when the wind speed increased, the wind field vortex was more noticeable, and the perturbation effect on the leaves was stronger. As a result, the probability that the abaxial surfaces of the leaves become exposed increases, which leads to greater coverage on the abaxial surfaces of the leaves. It was found that the airflow can create turbulence on the leaf surface, which causes the leaf to vibrate at a high frequency [32]. Wesely et al. (1985) also demonstrated that turbulence can have a significant impact on the deposition of small droplets [35]. Therefore, when the wind speed increases, the motion of the small droplets may also alter, which could account for the rise in the LDBE.

*3.3. Comparison in the Application Effect of ASS, KS, and SG*

The coverage of the ASS, KS, and SG at different parts of the canopy is shown in Table 7. The KS had high coverage on the adaxial surfaces of the leaves, with 94.3% of the upper leaves and 76.2% of the lower leaves, both of which produced significant differences ($p \leq 0.05$) from the other two applicators. This may be the result of the operator constantly swinging their arm and adjusting the angle of the applicator while operating. However, the effectiveness of the application depends on the operator's skill level [36], and the approach cannot be repeated. The ASS and SG also achieved more than 15% coverage in both parts, meeting the requirements for pest control [30]. Although the KS appears to have the best spraying effect, we also need to take droplet aggregation and pesticide loss into account [37,38]. Therefore, the actual control effect of the KS needs to be further verified.

**Table 7.** Coverage (%) and *DEAS* values of the three applicators in different parts of the canopy.

| Applicators | Leaves of Upper Canopy | | | Leaves of Lower Canopy | | |
|---|---|---|---|---|---|---|
| | Adaxial Surface | Abaxial Surface | *DEAS* (%) | Adaxial Surface | Abaxial Surface | *DEAS* (%) |
| ASS | 60.3 a | 25.8 b | 42.8 | 23.0 bc | 6.9 de | 29.7 |
| KS | 94.3 f | 17.1 g | 18.1 | 76.2 bd | 7.2 de | 9.5 |
| PB | 66.1 ag | 23.5 c | 35.5 | 36.5 b | 3.2 e | 8.7 |

Note: Different letters indicate a significant difference in the $p < 0.05$ level.

We must concentrate on the deposition on the abaxial surfaces of the leaves since powdery mildew typically appears on this part. For the upper canopy, the coverage rate was more than 15% for all three applicators. However, the KS had the lowest coverage of 17.1%, with the LDBE of 18.1%, indicating that the deposition effect on the abaxial surfaces of the leaves was subpar, while the ASS achieved the maximum coverage at a low spray volume due to the air assist. The abaxial surfaces of the lower leaves are the most difficult position for the droplets to deposit, and the three applicators' effects were not great on this part, with the coverage of all being less than 10%. The coverages of the KS and ASS were 7.2% and 6.8%, respectively, while the SG had the lowest coverage at 3.2%. Additionally, the wind field contributed to the strawberry air-assisted sprayer's higher deposition efficiency

on the abaxial surface, with a *DEAS* of 29.6%, while the KS and SG were 9.4% and 8.8%, respectively.

In conclusion, all three applicators were more efficient in depositing on the leaves of the upper canopy as well as the adaxial surfaces of the leaves in the lower canopy, and the coverage on the abaxial surfaces of the leaves in the lower canopy was inadequate. However, considering that high-volume spraying will increase the greenhouse's humidity, this may increase the risk of diseases such as downy and powdery mildew [39–41]. In addition, most greenhouses in northern China cannot adjust the humidity automatically. Thus, there are some advantages to the use of ASS, which had the lowest application volume at only 41.7% and 22.0% of the KS and SG, respectively.

In addition, the risk of exposure to operators must be taken into account, as exposure to excessive doses of pesticide can have a negative impact on their health, including the risk of cancer [42], neurological problems [43], and depression [44]. Since a greenhouse is a confined environment, operators may face a higher risk of exposure. According to a study by Nuyttens et al. (2003), using handheld equipment in greenhouse applications can result in a high level of exposure [45]. Therefore, using the KS and SG may pose a risk of exposure. Although studies have shown that wind delivery can reduce the exposure to some extent [46], the real exposure from using the ASS needs to be further validated.

## 4. Conclusions

In this study, an air-assisted strawberry sprayer for high-monopoly strawberry greenhouse application was constructed, and its performance was evaluated. The static and dynamic wind fields of the air-assisted spray unit of the sprayer were tested. In addition, the effects of the wind speed and spraying angle on the application effect were investigated, as well as the deposition on the different parts of the strawberry canopy. In addition, a comparison with the knapsack sprayer and spray gun for the deposition effect was conducted.

The results suggest that the presence of the jet angle caused a difference between the center and edge wind fields in the static wind field, and the difference was correlated with the distance and outlet wind speed. In the measurement of the dynamic wind field, we found that during the movement, the wind field rolled backward and produced an inward vortex, which may have been positively correlated with the wind speed and negatively correlated with the movement speed. The vortex can help us disturb the canopy and improve the coverage of the leaves in practice. In the study of the deposition pattern, we found that a spray angle of 30° had a high deposition efficiency on the abaxial surfaces of the leaves, accompanied by good coverage of both sides of the leaves. When the spray angle increased, the coverage of the adaxial surfaces of the leaves became higher, while the abaxial surfaces became lower. Moreover, the wind speed at the outlet also had an effect on the deposition. When the wind speed increased, the coverage on the abaxial surfaces of the leaves increased, with a decrease for the adaxial surfaces. Therefore, we should select the right application parameters based on the actual needs in practice. In the comparison of the ASS, KS, and SG, it was discovered that all three applicators had coverage of more than 15% on the leaves of the upper canopy as well as the adaxial surfaces of the leaves in the lower canopy. However, the coverages on the abaxial surfaces of the lower leaves were all below 10%. In addition, the ASS has the highest deposition efficiency on the abaxial surface among the three applicators, and its *DEAS* for the upper and lower canopies were 42.8% and 29.7%, respectively.

However, it should be noted that the coverage of the ASS on the abaxial surfaces of the leaves in the lower canopy was less than 10% for all spray angles and wind speeds, which needs to be improved in subsequent studies. The effectiveness of the spray against diseases during operation also needs to be evaluated as well. In addition, since there was no direct measurement of the application's exposure in the study, only references to other research findings, we also plan to test the exposure of the ASS and compare it with other applicators in subsequent studies.

**Author Contributions:** Conceptualization, J.S.; Methodology, J.S. and S.X.; Hardware, L.H. and X.R.; investigation, Y.F., Y.J. and S.X.; resources, J.S.; data curation, Y.F. and Y.Z.; writing—original draft preparation, S.X.; writing—review and editing, S.X., Y.F. and J.S.; project administration, S.X. and Y.F.; funding acquisition, J.S. All authors have read and agreed to the published version of the manuscript.

**Funding:** This research was supported by the China Agriculture Research System of the MOF and MARA (CARS-23-D02), as well as the administration of the Sanya Yazhou Bay Science and Technology City (Grant No. SYND-2022-23).

**Institutional Review Board Statement:** Not applicable.

**Data Availability Statement:** The data that support this study are available from the corresponding author upon reasonable request.

**Conflicts of Interest:** The authors declare no conflict of interest.

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
