# Peer review of "Evaluation of the Wind Field and Deposition Effect of a Novel Air-Assisted Strawberry Sprayer"

_agriculture, doi:10.3390/agriculture13020230_

Round 1
Reviewer 1 Report
Title: Evaluation of the wind field and deposition effect of a novel air- assisted strawberry sprayer
The study titled ‘Evaluation of the wind field and deposition effect of a novel air- assisted strawberry sprayer ’ emphasizes on lab and field evaluation of an air-assisted sprayer, designed to improve spray performance in strawberries. The inferior adaxial leave deposition and coverage had been an issue with most of the conventional applicators. Therefore, this study aims to tackle a crucial issue. Saying this, my major concern is with the poor experimental design of this study. The study was performed on very limited plants (no information available in the manuscript on the exact number ). Additionally, data have been collected in the small block without any replication. Therefore, the result of the study is expected to be biased in terms of plant density variation, spray conditions, plant height etc. The second issue is with the adopted methodology. Water-sensitive papers have been used for spray analysis. Firstly, WSPs data are reportedly unreliable for saturated spraying or in the condition of small droplets. WSP only provides the qualitative spray coverage value. For quantitative evaluation, actual deposition analysis is needed that involves a fluorometry analysis of dye collected on a different type of collectors. I would suggest authors to include deposition data for accurate comparative analysis of the sprayers. Lastly, the language of the manuscript is hard to follow. A major improvement is needed toward the language and readability of the paper. Following are my specific comments.
1. Consider using abaxial and adaxial surfaces of leaves instead of term upside and backside of leaves. These term sound scientific.
2. Line 60. Ramos is not the only author of this study, use Ramos et al., 2000. Modify whole manuscript accordingly.
3. At several instances (LINE 64-66, 77-80, 130-134), the use of commas (,) is unnecessary and wrong. For example, ‘Therefore, researchers have developed different applicators to improve the efficiency of strawberry application, vertical boom sprayers are the most common alternative’. The second comma is inappropriate, instead, you can easily break the sentences in two to improve readability. Many sentences are too long, try to write small and clear sentences. Modify the whole manuscript for such error
4. Line 62. Remove And before with.
5. Line 79 What is a frequency converter? Do you mean a potentiometer to adjust the voltage?
6. Section 2.1. The specification of the nozzle is not included, please consider providing the detail on the manufacturer, model, and type of nozzle.
7. Line 97. Clarify? do you mean evaluate?
8. Line 105. We can? Or you did? Try to use past tense instead of present in m&m section, for instance, ‘We obtained the wind speed…..’ or the wind speed was obtained in the UV plane. Similarly, anemometer was located (Line 107). Modify the whole section.
9. Line 186-193, 207-212. These are not results. This is your method and needs to move in the M&M section. The Results and Discussion section should solely focus on data presentation and discussion around the finding.
1 Table 3. Wind Speed is a standard term and can’t be used as a Difference between middle and edge wind speed(m⸱s-1). Why did you evaluate this difference? Is there any standard to justify this?
1 The major goal of your study was to improve the adaxial coverage however, looking at the results from tables 5&6, the difference between abaxial and adaxial coverage is still very significant (3-6 times). How will you justify this ?
Author Response
Thanks very much for your comment. Our reply for the comment is attached.

Reviewer 2 Report
The authors raised an interesting technical issue. Effective coverage of the leaves of sprayed plants from the underside is important in any crop not only in China. Therefore, I think the solution is innovative. The results focus on the technical part, which is undoubtedly very important. However, it is very interesting to see what the effectiveness of the spray was from the biological side. Perhaps the authors have results on the effectiveness of protection against agrophages after applying this innovative technique.
Author Response

(The authors gave the same response as above.)

Reviewer 3 Report
Dear Editor.
I have found a very interesting paper, I suggest that the authors include some improvements in the manuscript I have left some comments in the papers.
In addition, more information about the design of the new sprayer should be included.
The introduction should be improved a little with data of the crop at world level.
improve some figures.
to emphasize that this type of work is not carried out very often, hence the importance of this research.

Author Response

(The authors gave the same response as above.)

Round 2
Reviewer 1 Report
There has been a significant improvement in the manuscript quality however there is still a scope to improve the readability of the paper. For instance, the information on the spray replication needs to be included in the text, and the year needed to be added to the citation (e.g., Ramos et al., 2000). Also, please go through comment 8 from the last revision again and try to further modify the M&M. Finally, in the future, please highlight any change you make in the revised version, it was very hard for me to follow on the changes.
Author Response

(The authors gave the same response as above.)

Reviewer 3 Report
Dear Editor
the authors have included my suggestions.
Regards
Author Response
We really appreciate your recognition.And thanks for your insightful comments and suggestions on our manuscript.